# Androgen receptor binding sites enabling genetic prediction of mortality due to prostate cancer in cancer-free subjects

Shuji Ito[1,2,3], Xiaoxi Liu[1], Yuki Ishikawa [1], David D. Conti[4], Nao Otomo[1,5], Zsofia Kote-Jarai[6], Hiroyuki Suetsugu[1,7], Rosalind A. Eeles [6,8], Yoshinao Koike [1,9], Keiko Hikino [10], Soichiro Yoshino[1,7], Kohei Tomizuka[1], Momoko Horikoshi[11], Kaoru Ito [12], Yuji Uchio[3], Yukihide Momozawa [13], Michiaki Kubo[14], The BioBank Japan Project[15]*, Yoichiro Kamatani [15], Koichi Matsuda [16,17], Christopher A. Haiman[4], Shiro Ikegawa[2], Hidewaki Nakagawa [18] & Chikashi Terao [1,19,20] ✉

Prostate cancer (PrCa) is the second most common cancer worldwide in males. While strongly warranted, the prediction of mortality risk due to PrCa, especially before its development, is challenging. Here, we address this issue by maximizing the statistical power of genetic data with multi-ancestry meta-analysis and focusing on binding sites of the androgen receptor (AR), which has a critical role in PrCa. Taking advantage of large Japanese samples ever, a multi-ancestry meta-analysis comprising more than 300,000 subjects in total identifies 9 unreported loci including *ZFHX3*, a tumor suppressor gene, and successfully narrows down the statistically finemapped variants compared to European-only studies, and these variants strongly enrich in AR binding sites. A polygenic risk scores (PRS) analysis restricting to statistically finemapped variants in AR binding sites shows among cancer-free subjects, individuals with a PRS in the top 10% have a strongly higher risk of the future death of PrCa (HR: 5.57, $P = 4.2 \times 10^{-10}$). Our findings demonstrate the potential utility of leveraging large-scale genetic data and advanced analytical methods in predicting the mortality of PrCa.

Prostate cancer (PrCa) is the most common cancer in Europe and North America and the second most common cancer worldwide in males, accounting for an estimated 6.7% of cancer mortality in males[1]. Given its high mortality, the prediction of incidence and death due to PrCa would be of great interest from a public health perspective as implementing early detection and intervention for individuals with high prospective risk could be beneficial to both patients and health providers. To construct such predictive models, genetic components would be excellent sources as PrCa is evidenced to have a heritability of up to 58%, which is the highest among all cancers[2]. Family history has been utilized to identify the at-risk subjects, but this alone is not

sufficient for precise risk stratification. More recently, polygenic risk scores (PRS) based on genetic variants identified from genome-wide association studies (GWAS) of PrCa[3–14] have been developed[14–21]. However, prediction of the incidence of PrCa has limited clinical utility as PrCa could be latent, and autopsy studies showed a high prevalence of asymptomatic PrCa in older men. Predicting death due to PrCa will provide more value for clinical management but remains to be investigated.

Compared with conventional PRSs based on variants of GWAS at different *P* value thresholds, it has been shown that PRSs based on finemapped variants or variants with functional relevance to diseases

A full list of affiliations appears at the end of the paper. *A list of authors and their affiliations appear at the end of the paper. ✉e-mail: chikashi.terao@riken.jp

have a superior predictive performance[22,23]. There are two methods to identify variants that are informative for the PRS aside from the simple expansion of the GWAS. One is a statistical fine-mapping which may pinpoint potentially causal variants from the implicated association regions. Another is to prioritize variants in cell-type-specific regulatory elements relevant to the target phenotype. Considering the nature of PrCa as a male-specific cancer, in the current study we focused on the androgen receptor (AR), an important factor of PrCa development and progression and also a therapeutic target of PrCa[24]. AR is a transcription factor. Upon the binding of the active androgen dihydrotestosterone[25,26], AR is translocated into the nucleus and binds to hormone response elements of DNA and subsequently regulates the expression of various genes related to proliferation and differentiation[27]. AR was overexpressed and dysregulated in 56% of primary lesions and almost all metastatic lesions of PrCa[28]. Thus, we hypothesized that focusing on statistically finemapped variants within the AR-binding sites as putative causal variants may improve the prediction accuracy for incidence and death of PrCa, which could be useful in clinical settings.

Since European population is still the major source of genetic association studies in PrCa[14], non-European population would be useful to find unreported associations. While Japanese has relatively low prevalence of PrCa in comparison with European populations and African Americans[29], the previous studies showed substantial genetic overlap among populations[14].

In the present study, we conducted a multi-ancestry meta-analysis for PrCa and identified 171 loci associated with PrCa including 9 unreported loci. Furthermore, the fine-mapping analysis showed that variants with high posterior probability were enriched in AR-binding sites and PRS based on these variants predicts the PrCa mortality in cancer-free subjects. These findings provide insights into the basics underlying PrCa and clues for genetic prediction of the development and death of PrCa, resulting in potential early detection and therapeutic intervention for PrCa.

## Results

### A genome-wide association study of Biobank Japan

The overall study design of GWAS was shown in the Supplementary Fig. 1. First, we conducted a GWAS of Biobank Japan (BBJ) samples consisting of 8645 cases and 89,536 controls (Supplementary Table 1). We identified 32 significant loci including an unreported locus. The unreported signal at 16.q22.2-16q22.3 peaks at rs8052683, an intronic variant of tumor suppressor gene $ZFHX3$[30,31]. rs8052683 is located in an expression quantitative trait locus (eQTL) for $ZFHX3$ in the prostate (the risk allele of the variant lowering the expression of $ZFHX3$) in the GTEx data[32]. In line with the possible regulation of ZFHX3 expression by this variant, rs8052683 is positioned in the H3K27ac-marked region of the prostate[33].

We confirmed the relatively high SNP-heritability estimate of 26.2% (SE of 4.3%). As expected, we observed a strong genetic correlation of PrCa susceptibility between BBJ and Europeans (genetic effect correlation = 0.88 and $p = 0.36$ by popcorn software, indicating that genetic correlation is not different from 1, see "Methods"). Genetic correlation analyses also revealed significant positive correlations (FDR < 0.05) of PrCa with breast cancer (Supplementary Table 2, "Methods"). This correlation was also observed in Europeans (Supplementary Table 3). These findings are consistent with family studies that men with a family history of breast or prostate cancer had elevated prostate cancer risks[34–36]. Significant negative genetic correlations were found in cardiovascular-related phenotypes and similar trends were observed in Europeans (peripheral artery disease and chronic heart failure (Supplementary Table 3)), supported by enrichment of SNP heritability in the cardiovascular cell group (Supplementary Table 4, "Methods").

### Multi-ancestry meta-analysis identified 171 significant loci including nine unreported loci

Next, we conducted a multi-ancestry meta-analysis, using the results of BBJ and the summary statistics of a previous GWAS[14] for prostate cancer including European, African, and Hispanic ancestries assuming a random effect ("Methods", Table 1, Supplementary Fig. 2). The combined dataset consisted of 107,218 cases and 197,733 controls. A total of 6,720,553 variants were tested and 171 independent loci reached the genome-wide significance threshold ($\log_{10}$ [Bayes factor (BF)] > 6 and fixed-effect $P$ value $< 1 \times 10^{-5}$, for further details, see "Methods"). The 171 loci contained nine unreported loci PrCa including $ZFHX3$, $ARHGEF28$-$LINC01334$, and $GINS1$ regions (Table 1) which showed relevance to PrCa or functional mechanisms of variants on susceptibility to PrCa. rs4704108, located at the intergenic region between $ARHGEF28$ and $LINC01334$, is an eQTL for $ENC1$ in prostate tissues in the GTEx data[32] and in high LD with the lead eQTL SNP (rs17636369) of $ENC1$, suggesting that rs4704108 (or its tightly linked variant) is associated with PrCa via altering expression of $ENC1$ in the prostate. Risk allele of rs4704108 decreases expression of $ENC1$. rs11087515 is an intronic variant of $GINS1$ located at 20p11.21 which is expressed in high-grade prostate cancer and thus may be involved in the mechanism where cancer cells become invasive or metastatic[37].

### Fine-mapping by asymptotic Bayes factors

Taking advantage that different LD structures across different populations may enable higher resolution in pinpointing candidate causal variants, we further conducted fine-mapping analyses for all significant loci identified by the multi-ancestry meta-analysis ("Methods"). We calculated asymptotic Bayes factors (ABF) and built a credible set of putative causal variants with an incremental 95% probability which resulted in a total of 166 credible sets ("Methods" and Supplementary Data 1). The number of variants included in a given credible set was less than 10 in 127 (77%) credible sets (Fig. 1a), suggesting that we could finemap the majority of associations to a handful number of candidate causal variants.

To analyze the superiority of the multi-ancestry meta-analysis over the European-only analysis, we applied the same fine-mapping strategy to European data, restricting variants to those overlapped in the multi-ancestry meta-analysis (to harmonize conditions of fine-mapping). We found that the multi-ancestry meta-analysis resulted in a significantly less number of variants in credible sets compared with those in Europeans ($P = 1.34 \times 10^{-4}$, "Methods"). This indicates that multi-ancestry meta-analysis is useful to finemap association signals (to pinpoint or narrow down candidate causal variants) even though Europeans remain a major data source in the multi-ancestry meta-analysis (Fig.1b).

This fine-mapping strategy resulted in a total of 331 potentially causal variants with posterior probability (PP) > 0.1, among which 6 were coding variants (1.8%) including 5 missense variants (Supplementary Table 5). Out of 5 missense variants, rs2277283 with a PP of 0.99 is in $INCENP$ with scaled Combined Annotation-Dependent Depletion (CADD) score of 25.6. rs76832527 with a PP of 0.90 is the nonsynonymous variant of $ANO7$, which has been reported as a PrCa causal variant in a previous study[38]. rs138708 is a SNP of a unreported locus identified in the meta-analysis. The SNP is the nonsynonymous variant of $SUN2$ with a PP of 0.99 and a CADD score was 25.7, which is reported in a previous GWAS of Japanese[3].

### AR-binding sites as an important source of putative causal variants and construction of PRS based on AR-binding sites for potential clinical use

The small fraction (1.8%) of exonic variants among the candidate causal variants strongly suggest non-coding regions as primary sources. We hypothesized that AR-binding sites, hormone response elements for AR, are important to the susceptibility for PrCa incidence and

**Table 1 | The unreported significant loci for prostate cancer in the multi-ancestry meta-analysis**

| rsID | Chr | Position | Gene | Location | Ref | Alt | META | | | BBJ | | | EUR | | | AFR | | | HIS | | |
|---|---|---|---|---|---|---|---|---|---|---|---|---|---|---|---|---|---|---|---|---|---|
| | | | | | | | Beta | BF | Fixed P | Freq | Beta | P value | Freq | Beta | P value | Freq | Beta | P value | Freq | Beta | P value |
| rs2235558 | 1 | 24783311 | NIPAL3 | Intronic | A | G | 0.039 | 6.408 | 1.7E-08 | 0.747 | 0.064 | 8.63E-04 | 0.618 | 0.032 | 5.01E-05 | 0.747 | 0.062 | 1.04E-02 | 0.690 | 0.048 | 2.67E-01 |
| rs4893909 | 2 | 1.81E+08 | CWC22/SCHLAP1 | Intergenic | T | G | 0.043 | 7.110 | 3.10E-09 | 0.672 | 0.061 | 5.91E-04 | 0.729 | 0.043 | 1.43E-06 | 0.550 | 0.034 | 9.76E-02 | 0.698 | -0.007 | 8.67E-01 |
| rs4704108 | 5 | 7392666 | ARHGEF28/LINC01335 | Intergenic | G | A | -0.045 | 7.172 | 2.41E-09 | 0.214 | -0.086 | 1.69E-05 | 0.267 | -0.038 | 2.05E-05 | 0.383 | -0.028 | 1.83E-01 | 0.324 | -0.074 | 6.86E-02 |
| rs35055448 | 8 | 8135961 | TPD52/MIR5708 | Intergenic | G | A | -0.041 | 6.072 | 4.87E-08 | 0.543 | -0.064 | 1.74E-04 | 0.252 | -0.034 | 1.16E-04 | 0.153 | -0.045 | 1.32E-01 | 0.343 | -0.042 | 3.23E-01 |
| rs11259192 | 10 | 5726821 | FAM208B | 5' UTR | G | A | 0.062 | 7.217 | 3.24E-09 | 0.069 | 0.086 | 8.13E-03 | 0.119 | 0.054 | 2.38E-06 | 0.025 | 0.107 | 1.20E-01 | 0.163 | 0.135 | 8.72E-03 |
| rs11857866 | 15 | 33349563 | FMN1 | intronic | T | C | 0.043 | 7.282 | 2.47E-09 | 0.201 | 0.065 | 2.33E-06 | 0.310 | 0.039 | 1.66E-03 | 0.262 | 0.042 | 6.49E-02 | 0.208 | 0.056 | 2.55E-01 |
| rs8052683* | 16 | 73002421 | ZFHX3 | Intronic | G | A | -0.038 | 7.086 | 1.85E-07 | 0.253 | -0.113 | 4.46E-09 | 0.296 | -0.026 | 3.05E-03 | 0.185 | -0.033 | 2.31E-01 | 0.254 | -0.012 | 8.04E-01 |
| rs11087515 | 20 | 25399979 | GINS1 | Intronic | G | GA | 0.039 | 6.056 | 3.83E-08 | 0.105 | 0.054 | 6.09E-02 | 0.477 | 0.034 | 1.68E-05 | 0.470 | 0.062 | 4.45E-03 | 0.573 | 0.067 | 1.27E-01 |
| rs10154043 | 21 | 35535103 | MRPS6/LINCO0310 | Intergenic | C | T | -0.038 | 6.558 | 1.27E-08 | 0.229 | -0.047 | 1.66E-02 | 0.441 | -0.037 | 1.56E-06 | 0.536 | -0.060 | 3.23E-03 | 0.333 | 0.061 | 1.45E-01 |

Prioritized variants based on BF are indicated. Beta and BF of meta-analysis was calculated based on MANTRA algorithm. BBJ sample were analyzed with SAIGE software which contains a generalized linear mixed model (two-tailed).
Chr chromosome, Ref reference allele, Alt alternative allele, META meta-analysis, BBJ Biobank Japan, EUR Europeans, AFR Africans, HIS Hispanic, Beta beta of alternative allele, BF bayse factor, Fixed P value of fixed-effect meta-analysis using METAL software (two-tailed), Freq alternative allele frequency, UTR untranslated region.
*; Other variants in this gene region showed significant associations with p values smaller than 5E−08 in the fixed-effect model (Supplementary Fig. 3).

progression considering the critical roles of AR on PrCa development and AR as a therapeutic target in PrCa in clinical settings[39]. Indeed, *ZFHX3*, a unreported susceptibility gene identified in the study, was regulated by androgen in prostate cancer cells, and the relationship between androgen/AR signaling and ZFHX3 modulates prostate cancer development and progression[31].

First, we evaluated the enrichment of heritability of PrCa susceptibility in AR-binding sites of prostate tissues in common database. We computed LD scores using information of AR-binding sites in prostate tissues obtained from the ChIP-atlas[33] (this means that we used general AR-binding in prostate and did not match origins of data between PrCa susceptibility and AR binding). Then we conducted LD score regression using summary statistics of BBJ and Europeans and the computed LD scores (in combined with the basic model including common functional annotations as background, see "Methods"). For BBJ, 1.4% SNPs explained an estimated 46% SNP-heritability ($P = 6.1 \times 10^{-5}$ for enrichment analysis) (Supplementary Table 6); For Europeans, 1.3% SNPs explained an estimated 47% SNP-heritability ($P = 3.9 \times 10^{-9}$ for enrichment analysis). These findings indicate that susceptibility variants to PrCa are strongly enriched in AR-binding sites in a polygenic manner regardless of populations. In contrast, FOXA1 binding in prostate obtained from ChIP-atlas as control did not show heritability enrichment ($p = 0.075$, in BBJ).

Furthermore, we analyzed the enrichment of PrCa-associated variants in the AR-binding sites in normal prostate cells used for LDSC detected, using Genomic Regulatory Elements and Gwas Overlap algorithm (GREGOR)[40]. We found both lead variants identified in the multi-ancestry meta-analysis ($N = 171$) and putative causal variants with PP > 0.1 ($N = 331$) were significantly enriched in AR-binding sites ($P = 4.5 \times 10^{-26}$; fold enrichment 2.7; $P = 4.5 \times 10^{-53}$, fold enrichment 2.8, respectively). Notably, although the number of finemapped variants showing PP > 0.1 is nearly two times the number of lead variants, a slightly higher enrichment fold was observed, illuminating successful fine-mapping in the current study and the importance of the detection of critical functional annotations. We observed an enhanced fold enrichment of statistically finemapped variants with PP > 0.5 ($P = 1.3 \times 10^{-15}$, fold enrichment 3.1) suggesting the more credible the variants are, the stronger enrichment in AR-binding sites they show (Fig. 2 and Supplementary Table 7). We noticed that AR-binding sites detected in PrCa cells showed less enrichment for any groups of variants (Fig. 2 and Supplementary Table 7), suggesting an altered landscape of AR binding after the development of PrCa. These results were consistent with those when we used LD proxies of Asian populations (Supplementary Table 8).

Based on the importance of AR-binding sites on PrCa development shown above, we analyzed statistically finemapped variants in AR-binding sites as potentially causal variants. A total of 44 variants out of 331 statistically finemapped variants (PP > 0.1) were within AR-binding sites according to ChIP-atlas[33] ("Method", Supplementary Table 9). In the *IRX4* region, rs199577062 was the top SNP with a PP of 0.99. The variant is the lead eQTL for *IRX4* and changes the motif of AR-binding sites according to the JASPAR database (http://jaspar.genereg.net) (Fig. 3a). In the *RGS17* region, rs13215045 is the top SNP with a PP of 0.95 and has been reported as eQTL in the prostate. In the *FOXP4* region, rs1983891 is a SNP of the credible set with PP of 0.16, located at the AR-binding motif (Fig. 3b). The variant is not a significant eQTL for *FOXP4* but the expression of the risk allele increases the expression of *FOXP4*. rs12769682, rs12769019 and rs4962419 are SNPs with PP of 0.27, 0.16 and 0.15, respectively, are eQTL for *CTBP2*. The previous study indicated that *CTBP2* modulated the AR to promote prostate cancer progression[41]. rs1099399, well-known as a risk SNP for PrCa, was the top SNP with a PP of 1, which showed the result was well replicated. Taken together, the AR-binding sites of normal prostate cancer would be informative to pinpoint or narrow down putative causal variants for PrCa.

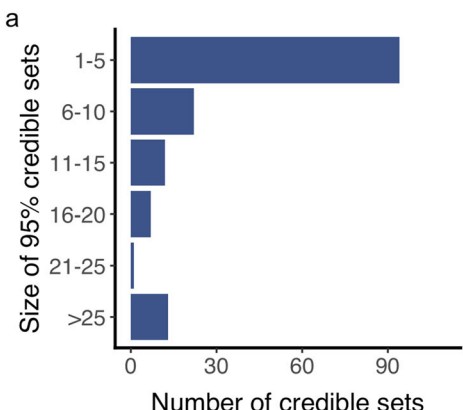

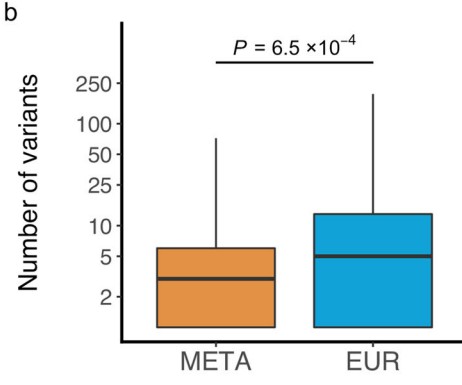

**Fig. 1 | Fine-mapping analysis. a** Number of 95% credible sets of potentially causal variants, binned by their sizes. **b** The comparison of the number of credible sets between the results of European and the multi-ancestry meta-analysis, $n = 118$ variants. Box plots indicate median (middle line), 25th, 75th, percentile (box) and 5th and 95th percentile (whiskers). $P$ value is shown (two sided Paired $t$-test). EUR, the result of Europeans; META, the result of the multi-ancestry meta-analysis. Source data are provided as a Source Data file.

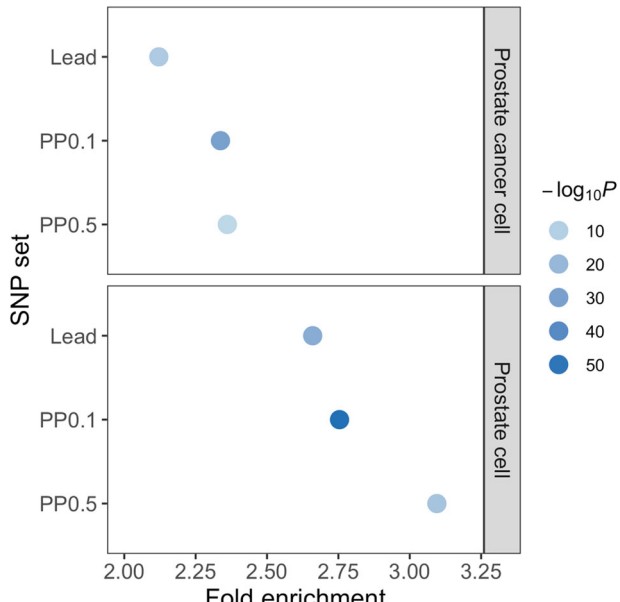

**Fig. 2 | GREGOR for androgen receptor (AR) binding sites.** Variants with high posterior probability (PP) were enriched in AR-binding sites in prostate cells. Fold enrichment and $P$ value are shown (see "Methods"). Lead, lead variants; PP0.1, the variants with PP > 0.1; PP0.5, the variants with PP > 0.5. Source data are provided as a Source Data file.

We analyzed whether PRSs could predict the death of PrCa in subjects without PrCa at baseline (registry of the BBJ, see "Methods"). We constructed a PRS using the results of the multi-ancestry meta-analysis which contained 6,720,152 SNPs, and the best parameters of each matrix were determined (See "Methods"). As a result, the model with $r^2$ of 0.1 and p value threshold of $1 \times 10^{-6}$, containing 952 SNPs, reached the highest AUROC (0.673; 95% confidence interval (CI), 0.664–0.682). We applied the PRS to the subjects who did not have PrCa or other cancers at the time of sample collection and analyzed follow-up data in the BBJ[42]. We performed the Cox proportional hazard model to assess the predictive value of the PRS for death from PrCa in the follow-up period. We observed that the PRS was significantly associated with PrCa death in a dose-dependent manner (HR:1.55, $P = 1.6 \times 10^{-7}$). The top 10% of subjects carrying high PRS showed HR of 3.52 ($P = 2.0 \times 10^{-5}$). These results were in line with the previous study in the UKB[21]. Considering the contribution of variants overlapping AR to

PrCa susceptibility motivated us to construct another PRS integrating AR information for variant selection using the lead SNPs and the SNPs within AR-binding sites (88,240 SNPs). As a result, the model with $r^2$ of 0.6 and p value threshold of $5 \times 10^{-4}$, containing 1107 SNPs, reached the highest AUROC (0.686; 95% CI, 0.676–0.695). We applied the AR-informed PRS to the same dataset as the analyses using the conventional PRS. As a result, we found that the PRS based on AR information showed better prediction in a dose-dependent manner (HR: 1.61, $P = 2.2 \times 10^{-10}$, Supplementary Table 10) than that by the conventional method. In addition, the top decile with high PRS also showed a strongly higher risk of the future death of PrCa in cancer-free subjects at registry of the BBJ (HR: 5.57, $P = 4.2 \times 10^{-10}$, Fig. 4 and Supplementary Table 10). We confirmed consistent results of better fitness of AR-informed PRS over normal PRS in additional analyses in which we avoided sample overlap between survival analyses and case-control studies (Supplementary Table 11 and "Methods"). In addition, we observed a trend of positive associations between AR-informed PRS and mortality in PrCa subjects (Supplementary Table 12). Taken together, these results suggest that AR plays fundamental and central roles on not only PrCa susceptibility, but on the future outcome of PrCa even in subjects without PrCa.

## Discussion

Here we present the PRS prioritizing AR-binding sites for potential clinical use based on large genetic fine-mapping study for the meta-analysis of PrCa comprising 107,218 cases and 197,733 controls in European, East Asian, African, and Hispanic ancestries.

We showed the good prediction ability of the PRS for the death from PrCa both in quantitative and qualitative manners. These findings were supported by strong heritability enrichment of PrCa susceptibility on AR-binding sites in normal prostate tissues obtained from public database and improved enrichment of statistically finemapped variants in GWAS significant signals. These indicate the genetic importance of AR-binding sites (in normal prostate tissues) shown in both GWAS significant and polygenic manners. Furthermore, the results indicate the SNPs within AR-binding sites in normal prostate tissues can be involved in not only the incidence of PrCa but also the progression of PrCa.

AR is a critical factor contributing to prostate cancer development and progression. Mutations in the AR gene have been discovered in prostate cancer, and their incidence may increase with tumor progression. On the other hand, Morova et al. demonstrate that AR-binding sites have a dramatically increased rate of mutations that is greater than any other transcription factors and specific to only

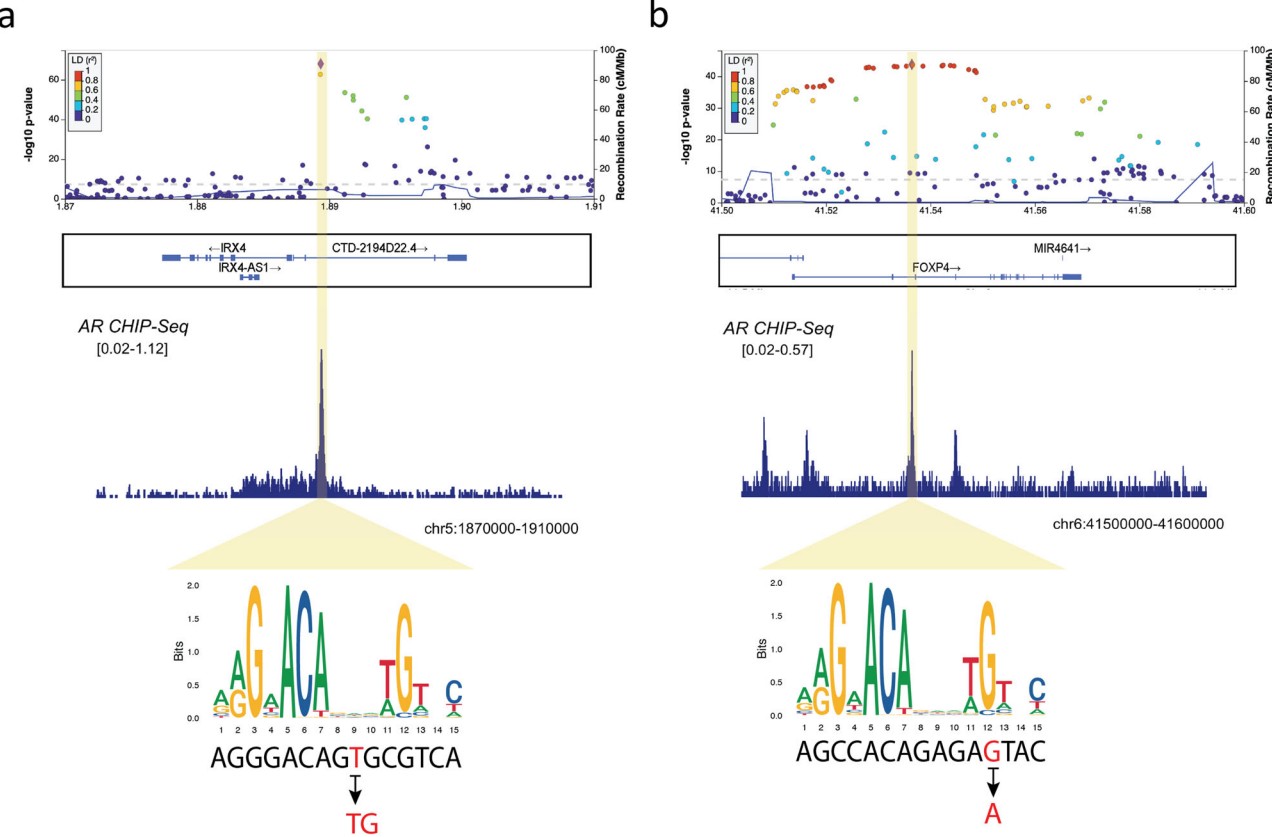

**Fig. 3 | Regional plot of the lead variants which affect androgen receptor binding sites. a** rs199577062. **b** rs1983891. Source data of the variants are provided as a Source Data file.

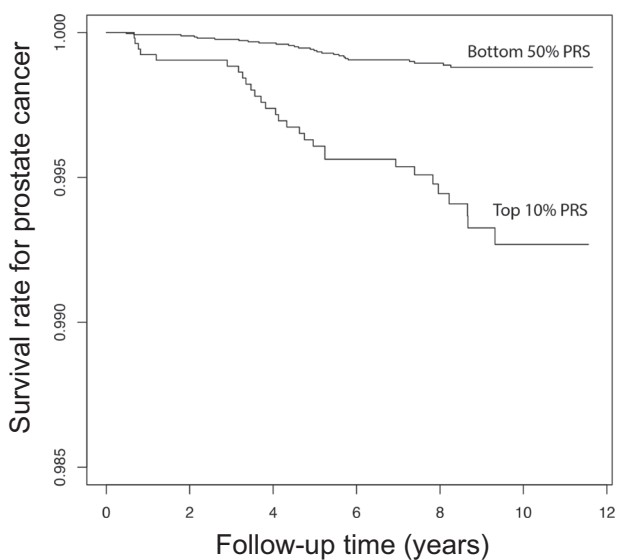

**Fig. 4 | Cox proportional hazard regression for death due to prostate cancer (PrCa) using polygenic risk scores (PRSs).** The comparison of survival rates between the top 10% of PRS and the bottom 50% is based on the variants located in androgen receptor binding sites. The x-axis indicates the time (years) from initial recruitment and the y-axis indicates the cumulative survival rate of PrCa. Hazard ratio and *P* value are shown in Supplementary Table 10.

prostate cancer, which can impact enhancer activity[43]. The alterations of the landscape of AR-binding sites in prostate cancer cells seem in line with decreased enrichment of statistically finemapped variants in AR-binding sites in prostate cancer cells in comparison with those in

normal prostate tissue. These results also suggest that AR-binding sites developing in prostate cancers are not genetically informative for prediction of PrCa or mortality of PrCa. It would be very interesting to compare AR-binding sites between normal prostate tissues and PrCa to characterize AR-binding sites in normal prostate tissue and understand molecular mechanisms underlying PrCa development. The association trend between AR-informed PRS and mortality was also observed in the analyses where we restricted to subjects with PrCa. Increased number of PrCa would confirm the association. We also noticed that landscape of AR-binding sites is quite distinct based on origins of tissues. When we analyzed AR bindings sites of white blood cells, we did not observe even a trend of heritability enrichment of PrCa susceptibility (data not shown), underscoring importance of selecting tissues as a source of functional variants to construct PRS which would be clinically useful.

In addition, we showed that multi-ancestry meta-analyses helped not only identify more association signals by increasing statistical power, but also narrow down candidate causal variants by taking advantage of different LD structures among different populations. Importantly, the addition of a relatively small number of different populations seemed to contribute to this function.

We also found nine unreported loci associated with PrCa and hundreds of statistically finemapped variants as potentially causal variants. In combination of AR-binding sites of normal prostate tissues, these variants would help us pinpoint causal genes and gain mechanistic insights into potentially causal variants and genes. The risk allele of rs8052683, located at the intronic region of *ZFHX3* which has a tumor suppressive role in PrCa[30], was found to be the same association direction across ancestries and showed associations with reduced expression in the prostate. rs4704108 is eQTL in prostate tissue for *ENC1*, one of the p53-induced genes[44,45]. rs11087515 was located at an

intronic region of *GINS1* expressing in high-grade prostate cancer[37] with potential roles for PrCa susceptibility[43,46,47].

Fine-mapping analyses identified several causal genes. *SUN2* is an inner nuclear membrane protein that plays a major role in nuclear-cytoplasmic connection by the formation of a 'bridge' across the nuclear envelope[48]. *SUN2* also interacts with lamins, functioning as a nuclear skeleton in the neoplasm. Dysregulation of SUN2, as a characteristic nuclear envelope protein, is associated with many human diseases, including cancers[49]. Notably, Yajun et al. reported loss of *SUN2* promoted the progression of prostate cancer by regulating fatty acid oxidation[50].

*PYGB* encodes glycogen phosphorylase which metabolizes glycogen and is reported as an upregulated gene in prostate cancer tissues. *PYGB* silencing suppressed the growth and promoted the apoptosis of prostate cancer cells by affecting the NF-κB/Nrf2 signaling pathway[51]. *IRX4*, a member of the Iroquois family of homeobox transcription factors predominantly expressed in the ventricle of the heart, could function as a tumor suppressor in the prostate via the vitamin D receptor pathway[52], which enhances antitumor immunity by inhibiting Wnt/β-catenin signaling[53].

Wu et al. confirmed that PAX5-induced upregulation of *FOXP4-AS1* and *FOXP4* contributed to tumorigenesis of PrCa[54].

Our study has several limitations. First, we did not include rare variants (MAF < 0.005) in our analyses. Such variants may play a crucial role in PrCa incidence. Further studies using whole-genome sequencing data are needed to elucidate the causality of rare variants and structural variants. Second, the majority of patients in our study were Europeans, resulting in a bias towards Europeans. As we showed, multi-ancestry meta-analyses contributed to narrowing down statistically finemapped variants. More non-Europeans are needed to further identify the true causal functional variants, which may lead to the elucidation of additional genetic factors. Third, due to limited number of follow-up data of PrCa subjects, we could not fully address AR-informed PRS and future death due to PrCa in PrCa subjects. Lastly, prediction of AR-informed PRS on mortality due to PrCa in cancer-free subjects and cancer cases should be addressed in European populations to show its generalizability, especially in a cohort specific to PrCa.

In summary, our large-scale multi-ancestry meta-analysis of GWASs provides further insights into the genetic architecture of PrCa susceptibility, and the unreported genetic loci identified may lead to the development of drug discovery and intervention for patients with PrCa. We demonstrated the potential utility of the PRS using candidate causal variants in AR-binding sites to predict the future mortality risk of PrCa.

## Methods

This research complies with all relevant ethical regulations and approved by The Ethics Review Committee in RIKEN under approval ID of 2021-19 and 17-17-16.

### Subjects

All cases and controls in the BBJ GWAS were collected in the BioBank Japan (https://biobankjp.org/english/index.html), which is a biobank that collaboratively collects DNA and serum samples from 12 medical institutions in Japan and recruited approximately 200,000 patients with a diagnosis of at least one of 47 diseases (including 13 cancers)[55]. We selected 8645 pathologically proven PrCa patients and 89,536 male subjects without PrCa as control from BBJ. We used all available samples we have and no statistical method was used to predetermine sample size. Since the target disease is prostate cancer, only male subjects were recruited and analyzed. We obtained informed consent from all participants by following the protocols approved by committees of the RIKEN Center for Integrative Medical Sciences and Institute of Medical Sciences at The University of Tokyo. We complied with all relevant ethical regulations[56].

### Genotyping and quality control

We genotyped samples with the Illumina HumanOmniExpressExome BeadChip or a combination of the Illumina HumanOmniExpress and HumanExome BeadChips.

For quality control of samples, we excluded those with: (1) a sample call rate of <0.98; and (2) outliers from East Asian clusters identified by a principal component analysis using genotyped samples. For quality control of genotypes, we excluded variants meeting any of the following criteria: (1) call rate <99%; (2) $P$ value for Hardy–Weinberg equilibrium (HWE) < $1.0 \times 10^{-6}$. We also excluded SNP with a large allele frequency difference between the reference panel and the samples (>0.06).

### Imputation

We utilized the 1000 Genomes Project Phase 3 (1KGP3; May 2013, $n = 2504$) and an in-house Japanese whole-genome sequence dataset obtained from 3,256 BBJ subjects (JEWEL 3 K) for imputation. We imputed genotype dosages with minimac4[57]. After imputation, we excluded variants with an imputation quality of Rsq <0.3 and MAF < 0.005.

### GWAS and meta-analysis

We conducted a GWAS of BBJ samples by applying a generalized linear mixed model using SAIGE (version 0.35.8.3)[58], which consisted of two steps. In Step 1, we fit a null logistic mixed model using genotype data, and the top 10 principal components (PCs) were incorporated as covariates. In Step 2, the single-variant association tests were performed by using the imputed variant dosages. We divided the datasets into two depending on the registration period and conducted the association studies separately to see the robustness of the observed association. We further performed a fixed-effects meta-analysis for these two datasets using METAL (two-tailed)[59].

### Multi-ancestry meta-analysis

We used summary statistics of PrCa GWAS divided into three ancestries (European, African and Hispanic)[14]. We excluded samples of East Asian ancestries due to duplicate samples. To account for the ancestral heterogeneity in each study, we applied the MANTRA algorithm in our multi-ancestry meta-analysis analysis[60]. We considered that a variant was significantly associated with PrCa when its $\log_{10}BF > 6$ and $P_{\text{Fixed effect}} < 1 \times 10^{-5}$ by using METAL (two-tailed) according to a previous simulation result[61]. We excluded variants with a heterogeneity score > 6.

### Fine-mapping

There are no fine-mapping methods currently to handle multiple populations. Furthermore, for some ethnic groups analyzed here, we have difficulty accessing a large reference panel to account for LD. Thus, we finemapped a Bayesian approach by adapting the method proposed by Maller et al. to assign a posterior probability of inclusion (PIP) to each variant and construct 95% credible sets[56].

For the multi-ancestry meta-analysis by using METAL, we defined a significantly associated locus of a lead variant as 1 Mb of its surrounding sequences in both directions. Then, we extended the region to nearby significant variants and their 1 Mb surrounding sequences as far as a significant variant was contained in the defined region. For each locus, we calculated asymptotic Bayes factors as previously described[62] which is a LD-independent method. Bayes factors can be approximated from summary statistics (such as p value and standard error of the effect size of each variant from GWAS) without individual-level genotype data. Then we defined the subset of SNPs based on posterior probability (PP), as 95% likely to contain the causal disease-associated SNP (https://github.com/chr1swallace/finemap-psa). These credible SNP sets were then annotated for putative function.

We further compared the number of variants in the credible sets between the results of the multi-ancestry meta-analysis and that of European alone. We restricted variants present in both the multi-ancestry meta-analysis and European results to harmonize conditions and conducted fine-mapping as described above for each of the multi-ancestry meta-analysis and European alone. Then we focused on the significant regions overlapped between the two results and compared the number of variants in the credible sets. The comparison was estimated by paired t-test using R (version 4.0.3).

### Heritability enrichment analyses and genetic correlations

We estimated the heritability of PrCa with BBJ GWAS results using linkage disequilibrium score regression (LDSC, version 1.0.0)[63]. We excluded variants in the human leukocyte antigen region (chromosome 6: 26–34 Mb). We further calculated heritability z-scores and standard errors (SEs) to assess the reliability of heritability estimation. The disease prevalence of PrCa used for the heritability estimations was 0.08 according to a previous report[64]. We also estimated genetic correlation of PrCa susceptibility between BBJ and EUR by popcorn software, using the sumstats of PrCa in EUR used for the meta-analysis. We additionally estimated genetic correlations with the 42 target diseases of the BBJ. We also evaluated the enrichment of the heritability of histone marks in 220 different cell types and 10 different tissue types.

### Annotation of the significant SNPs in our meta-analysis

We extracted histone modifications (H3K4me1 mark often found near regulatory elements, H3K4me3 mark often found near promoters, and H3K27ac mark often found near active regulatory elements on 7 cell lines) and DNase hypersensitive sites (DNaseI hypersensitivity clusters in 125 cell types) defined by ENCODE (version 3)[65], and, if SNPs are intergenic, enhancers (hg19) mapped by FANTOM5 (phase2.5) from the UCSC database[66,67]. We also used Combined Annotation-Dependent Depletion (CADD) score[68], a method integrating multiple annotations, to evaluate the functional potential of SNVs.

### Enrichment analysis of PrCa statistically finemapped variants in androgen receptor binding sites

To confirm the enrichment of AR-binding sites, we evaluated two methods, LDSC and GREGOR tool, a SNP-matching-based method to test for enrichment[69]. Regarding the information on AR-binding sites, we downloaded the bed file of AR-binding sites in the prostate from ChIP-atlas[33]. We extracted the bed files in normal prostate cells and prostate cancer cells, each file was evaluated for subsequent analysis.

For LDSC, we assessed heritability enrichment in AR-binding sites as previously described[70] for each of European and BBJ GWAS results. Taking ChIP-seq data of AR in normal prostate, we computed LD scores of AR binding in normal prostate using LD structure of East Asians and Europeans, respectively, and used them for LDSC. We used the BBJ sumstats with use of EAS LD scores and European population sumstats with European's LD scores for LDSC. We used the 53 basic model annotations (v1.0) to control inflation of the results as previously described[70]. We excluded variants within the major histocompatibility complex (MHC) region (chromosome 6: 25–34 Mb) from the regression analysis.

For GREGOR, we calculated the fold-enrichment expectation and an enrichment *P* value that represents the probability that the overlap of control SNPs represented as a cumulative probability distribution is greater than or equal to the observed overlap with PrCa potentially causal variants. We conducted the analysis using three SNP sets: lead variants identified in the meta-analysis, the variants with PP > 0.1, the variants with PP > 0.5. Reference based on European population was used in the analysis because the majority of the samples are from Europeans.

### PRS for prediction of susceptibility to PrCa

We conducted an ethnic-ancestry meta-analysis using the 1st BBJ, European, African, and Hispanic ancestries and determined the weight for the multi-ancestry PRS. Using these weights, we calculated the PRS for the 2nd BBJ and validated its performance. To determine the best parameter, we applied the pruning and thresholding method, the standard method of analysis, to construct the PRS. Specifically, we used the clump function of PLINK version 1.90 to generate eligible SNPs by setting a 250 kb window. We set seven linkage disequilibrium pruning thresholds ($r^2$) of 0.3, 0.4, 0.5, 0.6, 0.7, 0.8, and 0.9, and a total of 20 thresholds of p values in the GWAS ($5 \times 10^{-8}$, $5 \times 10^{-7}$, $1 \times 10^{-6}$, $5 \times 10^{-6}$, $1 \times 10^{-5}$, $5 \times 10^{-5}$, $5 \times 10^{-4}$, 0.005, 0.01, 0.05, 0.1, 0.2, 0.3, 0.4, 0.5, 0.6, 0.7, 0.8, 0.9, 1). As weights, we used the natural logarithms of the GWAS odds ratios (ORs) for PrCa. The SNP alleles used in the PRS were aligned to risk alleles for PrCa susceptibility. The PRS was the sum of the weighted allele counts (by their respective GWAS effect sizes) across all SNPs included in the PRS.

Next, we addressed the possible improvement of the performance of the PRS by prioritizing SNPs included in the predictive model. Since we showed significant heritability enrichment of PrCa in the AR-binding sites. We prioritized SNPs overlapping with AR-binding sites of the prostate and adopted the same manner for generating the PRS described above.

We further investigated the potential of integrating functional annotations to improve the predictive performance and multi-ancestry portability of the PRS. We hypothesized such improvement might be achieved by prioritizing SNPs located within the AR-binding sites, given the significant enrichment of AR sites in the heritability of PrCa. To test this hypothesis, we repeated the PRS analysis while restricting the variants that overlapped with the AR-binding peaks (determined by CHIP-Seq analysis).

### Survival analysis of death due to PrCa in cancer-free subjects

We used the BBJ follow-up data in which ~142 K participants were followed up to 10 years after BBJ registration to monitor their survival. Causes of death (coded with ICD10 codes) were recorded by accessing the national vital registration system used for input survey of medical and social welfare at the Ministry of Health, Labour and Welfare of the Japanese Government. We restricted subjects to males and not affected by any cancers at the registry, resulting in a total of 54,033 male subjects. We focused on associations between the death of PrCa and PRS and applied the Cox Proportional hazard model with top 10 PCs, age, smoking, basic disease status, and SNP array type as covariates. We evaluated fitness of the PRS on survival/death of PrCa in a quantitative manner (treating PRS as a quantitative value) or in a qualitative manner (taking the top 10% of PRS as cases and the below 50% as controls).

To confirm the findings, we conducted additional analyses in which we avoid sample overlap between survival analyses and case-control study (Supplementary Fig. 4). We split the BBJ controls into two sets based on availability of follow-up information. We reconducted association analyses using the 1st set of subjects without follow-up data and meta-analyzed the association results in a multi-ancestry manner. We then reconstructed PRS using the same variants with different beta coefficients. For intuitive interpretation, we used the same parameters for PRS as the final model as written above ($r^2$ of 0.6 and p value threshold of $5 \times 10^{-4}$) and conducted survival analyses using the 2nd set of control subjects in Cox proportional hazard model with the same covariates as above.

### Survival analysis in PrCa subjects

To evaluate an association between AR-informed PRS and mortality in subjects with PrCa, we conducted survival analyses in case subjects. We used the same AR-informed PRS described in the former part of the

previous section. In this analysis, we took mortality due to any causes as outcome. We used the same covariates described above and analyzed the association in Cox proportional hazard model.

## Statistics & reproducibility

No statistical method was used to predetermine sample size because we used all available case samples we have to maximize statistical power.

## Reporting summary

Further information on research design is available in the Nature Portfolio Reporting Summary linked to this article.

## Data availability

The GWAS and Meta-analysis summary statistics generated in this study is available in the JENGER database and GWAS catalog (BBJ: http://ftp.ebi.ac.uk/pub/databases/gwas/summary_statistics/GCST90269001-GCST90270000/GCST90269956j, A multi-ancestry meta-analysis: http://ftp.ebi.ac.uk/pub/databases/gwas/summary_statistics/GCST90269001-GCST90270000/GCST90269957). The BBJ Genotype data is available JGAS000114 in NBDC database. The authors are not allowed to deposit individual disease affection status and survival data in common database. Source data are provided with this paper. The remaining data are available within the article, Supplementary Information or Source Data file. Source data are provided with this paper.

## Code availability

The code of statistical analyses is available on GitHub (https://github.com/Shuji2022/PrCa_GWAS) and is also archived in Zenodo (https://zenodo.org/badge/latestdoi/633776153).

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

## Acknowledgements

The sample and data used for this study were provided by the BBJ supported. by the Japan Agency for Medical Research and Development (AMED) Grant JP21ek0109555, JP21tm0424220 and JP21ck0106642 (CT), Japan Society for the Promotion of Science (JSPS) KAKENHI Grant JP20H00462 (CT).

## Author contributions

Conceptualization: R.A.E., C.A.H., C.T. Methodology: S.I., X.L., Y.I., D.D.C., N.O., Z.K.-J., H.S., R.A.E., Y. Koike, K.H., S.Y., K.T., C.T.. Investi-gation: S.I., D.D.C., Z.K.-J., R.A.E., C.A.H., C.T.. Visualization: S.I., X.L., Y.I., C.T. Funding acquisition: C.T.. Project administration: C.T.. Supervision: C.T. Writing – original draft: S.I., X.L., Y.I., C.T. Writing – review & editing: S.I., X.L., Y.I., D.D.C., N.O., Z.K., H.S., R.A.E., Y. Koike, K.H., S.Y., K.T., M.H., K.., Y.U., Y.M., M.K., Y. Kamatani, K.M., C.A.H., S.I., H.N., C.T.

## Competing interests

The authors declare no competing interests.

## Additional information

[1]RIKEN Center for Integrative Medical Sciences, The Laboratory for Statistical and Translational Genetics, Yokohama, Japan. [2]RIKEN Center for Integrative Medical Sciences, The Laboratory for Bone and Joint Diseases, Yokohama, Japan. [3]Department of Orthopedic Surgery, Shimane University, Izumo, Japan. [4]Center for Genetic Epidemiology, Department of Population and Public Health Sciences, Keck School of Medicine, University of Southern California, Los Angeles, CA, USA. [5]Department of Orthopedic Surgery, School of Medicine, Keio University, Tokyo, Japan. [6]The Institute of Cancer Research, London, UK. [7]Department of Orthopedic Surgery, Graduate School of Medical Sciences, Kyushu University, Fukuoka, Japan. [8]Royal Marsden NHS Foundation Trust, London, UK. [9]Department of Orthopedic Surgery, Hokkaido University Graduate School of Medicine, Sapporo, Japan. [10]RIKEN Center for Integrative Medical Sciences, The Laboratory for Pharmacogenomics, Yokohama, Japan. [11]RIKEN Center for Integrative Medical Sciences, The Laboratory for Genomics of Diabetes and Metabolism, Yokohama, Japan. [12]RIKEN Center for Integrative Medical Sciences, The Cardiovascular Genomics and Informatics, Yokohama, Japan. [13]RIKEN Center for Integrative Medical Sciences, The Laboratory for Genotyping Development, Yokohama, Japan. [14]Haradoi Hospital, Fukuoka, Japan. [15]Laboratory of Complex Trait Genomics, Graduate School of Frontier Sciences, The University of Tokyo, Tokyo, Japan. [16]Institute of Medical Science, The University of Tokyo, Laboratory of Genome Technology, Human Genome Center, Tokyo, Japan. [17]Graduate School of Frontier Sciences, The University of Tokyo, Laboratory of Clinical Genome Sequencing, Department of Computational Biology and Medical Sciences, Tokyo, Japan. [18]RIKEN Center for Integrative Medical Sciences, Laboratory for Cancer Genomics, Yokohama, Japan. [19]Shizuoka General Hospital, The Clinical Research Center, Shizuoka, Japan. [20]School of Pharmaceutical Sciences, University of Shizuoka, The Department of Applied Genetics, Shizuoka, Japan.
✉e-mail: chikashi.terao@riken.jp

## The BioBank Japan Project

**Akihide Masumoto[21], Akiko Nagai[22], Daisuke Obata[23], Hiroki Yamaguchi[24], Kaori Muto[22], Kazuhisa Takahashi[25], Ken Yamaji[26], Kozo Yoshimori[27], Masahiko Higashiyama[28], Nobuaki Sinozaki[29], Satoshi Asai[30,31], Satoshi Nagayama[32], Shigeo Murayama[33], Shiro Minami[34], Takao Suzuki[29], Takayuki Morisaki[35], Wataru Obara[36], Yasuo Takahashi[31], Yoichi Furukawa[37], Yoshinori Murakami[38], Yuji Yamanashi[39] & Yukihiro Koretsune[40]**

[21]Iizuka Hospital, Fukuoka, Japan. [22]Department of Public Policy, Institute of Medical Science, The University of Tokyo, Tokyo, Japan. [23]Center for Clinical Research and Advanced Medicine, Shiga University of Medical Science, Shiga, Japan. [24]Department of Hematology, Nippon Medical School, Tokyo, Japan. [25]Department of Respiratory Medicine, Juntendo University Graduate School of Medicine, Tokyo, Japan. [26]Department of Internal Medicine and Rheumatology, Juntendo University Graduate School of Medicine, Tokyo, Japan. [27]Fukujuji Hospital, Japan Anti-Tuberculosis Association, Tokyo, Japan. [28]Department of General Thoracic Surgery, Osaka International Cancer Institute, Osaka, Japan. [29]Tokushukai Group, Tokyo, Japan. [30]Division of Pharmacology, Department of Biomedical Science, Nihon University School of Medicine, Tokyo, Japan. [31]Division of Genomic Epidemiology and Clinical Trials, Clinical Trials Research Center, Nihon University. School of Medicine, Tokyo, Japan. [32]The Cancer Institute Hospital of the Japanese Foundation for Cancer Research, Tokyo, Japan. [33]Tokyo Metropolitan Geriatric Hospital and Institute of Gerontology, Tokyo, Japan. [34]Department of Bioregulation, Nippon Medical School, Kawasaki, Japan. [35]Division of Molecular Pathology IMSUT Hospital, Department of Internal Medicine Project Division of Genomic Medicine and Disease Prevention The Institute of Medical Science The University of Tokyo, Tokyo, Japan. [36]Department of Urology, Iwate Medical University, Iwate, Japan. [37]Division of Clinical Genome Research, Institute of Medical Science, The University of Tokyo, Tokyo, Japan. [38]Department of Cancer Biology, Institute of Medical Science, The University of Tokyo, Tokyo, Japan. [39]Division of Genetics, The Institute of Medical Science, The University of Tokyo, Tokyo, Japan. [40]National Hospital Organization Osaka National Hospital, Osaka, Japan.

