## [Peer Review File · Nature Communications]

Androgen receptor binding sites enabling genetic prediction of mortality due to prostate cancer in cancer-free subjectsREVIEWER COMMENTS

Reviewer #1 (Remarks to the Author): expert in prostate cancer genomics

The authors addressed the challenges of prediction of prostate cancer (PrCa) mortality risk by maximizing the statistical power of genetic data with trans-ancestry meta-analysis and focusing on binding sites of the androgen receptor (AR), which has a critical role in PrCa in a large Japanese cohort by conducting a trans-ethnic meta-analysis comprising more than 300,000 subjects. Their analyses identified 9 novel loci including ZFX3, a tumor suppressor gene. The authors stated that the trans-ancestry meta-analysis narrowed down these candidate causal variants, enriched in AR binding sites compared to European-only studies. A polygenic risk scores (PRS) analysis of candidate causal variants in AR binding sites showed among cancer-free subjects. Their results showed that individuals with a PRS in the top 10% have a higher risk of the future death of PrCa (HR: 5.57, $P = 4.2 \times 10^{-10}$).

This manuscript is of significance to prostate cancer epidemiology. . There is recent literature surrounding the calculation of a polygenic risk score in prostate cancer. . The novel component appears to relate to the method for finding significant variants specific to the androgen receptor (AR) using trans-ancestry meta-analysis in a large Japanese population.

Select related literature:

Chen, F.; Darst, B.F.; Madduri, R.K.; Rodriguez, A.A.; Sheng, X.; Rentsch, C.T.; Andrews, C.; Tang, W.; Kibel, A.S.; Plym, A., et al. Validation of a multi-ancestry polygenic risk score and age-specific risks of prostate cancer: A meta-analysis within diverse populations. *Elife* 2022, 11, doi:10.7554/eLife.78304.

Reference #14 in manuscript:

Conti DV, Darst BF, Moss LC, Saunders EJ, Sheng X, Chou A, Schumacher FR, Olama AAA, Benlloch S, Dadaev T, Brook MN, Sahimi A, Hoffmann TJ, Takahashi A, Matsuda K, Momozawa Y, Fujita M, Muir K, Lophatananon A, Wan P, et al. 2021. Trans-ancestry genome-wide association meta-analysis of prostate cancer identifies new susceptibility loci and informs genetic risk prediction. *Nature Genetics* 53:65–75. DOI: <https://doi.org/10.1038/s41588-020-00748-0>, PMID: 33398198

Plym A, Penney KL, Kalia S, Kraft P, Conti DV, Haiman C, Mucci LA, Kibel AS. 2022. Evaluation of a multiethnic polygenic risk score model for prostate cancer. *Journal of the National Cancer Institute* 114:771–774. DOI: <https://doi.org/10.1093/jn93ci/djab058>, PMID: 337926

I have a concern about this work supporting one of the conclusions of this manuscript. The authors claimed that their results support the androgen receptor (AR) plays fundamental and central roles not only PrCa susceptibility but on the future outcome of PrCa even in subjects without PrCa (Lines 247-248) and in "the potential early detection and therapeutic intervention for PrCa" (Line 104). This is in contradiction to Lines 281 to 284, where the authors stated that their analysis of AR binding sites in leukocytes, did not result in heritability enrichment of PrCa susceptibility. Thus, selecting prostate cancer tissue would be better to construct PRS. This finding decreases the utility of PRS in AR being used to predict PrCa in individuals who do not have PrCa. A better accompaniment to screening in PrCa prevention would be a blood test instead of an invasive biopsy of PrCa tissue which a clinician would not obtain a prostate tissue sample unless a patient had already been diagnosed or was being evaluated for PrCa.

Additionally, I believe that the inclusion of a Japanese cohort to calculate polygenic risk scores lacks supportive documentation explaining why this population is an ideal population to study the genetics of prostate cancer. Why was the BBJ cohort selected? What is the incidence, morbidity and mortality of prostate cancer in Japanese men and how does it compare to African American men who carry a disproportionate share of prostate cancer burden. The authors should revise this manuscript to provide more documentation supporting their selection of this cohort.

I do not detect any flaws in the data analysis. I believe the methodology is sound and meets the expected standards in this field of study. There appears to be enough detail provided in the

methods to reproduce the work reported in this manuscript.

Reviewer #2 (Remarks to the Author): expertise in GWAS bioinformatics

This study conducts a meta-analysis of genome-wide association studies of prostate cancer, identifies 9 novel loci, creates a polygenic risk score (PRS) for prostate cancer and tests for association with prostate cancer death. Although the study is large and brings some insight into the genetic etiology of prostate cancer, it is limited by having only an incrementally larger dataset than the previous multi-ancestry GWAS and lacks an independent sample set for evaluating polygenic risk scores. Specific comments are below.

1) This study takes advantage of data from Biobank Japan (BBJ) to increase diversity; however, the added value is somewhat limited by the fact that the study excluded the East Asian cases and controls that were included in the previous multi-ancestry GWAS meta-analysis from prostate cancer. Thus, the number of EAS is about the same, although the number of EAS controls is larger. The authors excluded the EAS subjects from the previous study, because some of them overlapped with BBJ. This is a reasonable approach; however, the authors could have excluded the Biobank Japan subjects from the EAS summary statistics of the previous GWAS using the MetaSubtract R package and then included the non-overlapping subjects in the analysis.

2) The author identified 171 independent loci (including 9 novel loci). However, no criteria are given as to how the authors determined that the loci were independent. The analysis method used is also not clear. The authors mention using MANTRA in the methods, meta-analysis assuming random effects on line 129 and fixed effects meta-analysis on line 132. In addition, instead of adopting the standard genome-wide p-value threshold of $p < 5 \times 10^{-8}$, the authors use a slightly different significance threshold based on \log_{10} Bayes Factor > 6 and fixed effects P-value $< 1 \times 10^{-5}$. In other studies that have employed these criteria, the loci have not always replicated in subsequent studies. Most of the novel loci that the authors report (except for one locus) have a fixed effects meta-analysis p-value $< 5 \times 10^{-8}$. Given that the one novel locus with a fixed effects $p > 5 \times 10^{-8}$, is driven primarily by BBJ (with little or no association in other ancestries), it does make one question whether this is indeed a real locus. Are there other signals in the region supporting this association in LocusZoom plots? Are the findings replicated in another study, such as in the EAS subjects from the previous multi-ancestry analysis that were not included in this study? In the absence of replication, maybe it would be prudent to classify this locus as suggestive (not genome-wide significant).

3) The authors test the association between the PRS and the risk of prostate cancer death among a cohort of men who do not have prostate cancer at baseline in BBJ. However, this analysis is problematic in that the authors appear to have used the same men included in the discovery and creation of PRS to test for the association with prostate cancer mortality. Although one may argue that the endpoint here is prostate cancer mortality and not incidence, in order to die from prostate cancer, one must first be diagnosed with prostate cancer. The men who die are just a subset of those who were diagnosed with prostate cancer. They are not an independent set of men, and mortality is not independent outcome. Moreover, the more important question to ask is whether the PRS is predictive of death among men who have already been diagnosed with prostate cancer. Clinically, this is more useful than predicting prostate cancer death among those who do not have prostate cancer yet. Other prostate PRS have been shown to predict prostate cancer death in a cohort of healthy men, because prostate PRS are predictive of prostate cancer incidence (which is necessary for prostate cancer death). Finally, it is not clear which PRS are presented in the paper. The authors discuss testing several LD pruning thresholds in the methods and investigating the integration of functional annotations, but do not state which LD threshold was ultimately used OR provide the variants/weights used in either PRS presented (e.g., the PRS without inclusion of AR information and the PRS with inclusion of AR information). Both PRS were also only tested in EAS, so their performance in other ancestries is unknown. Hazard ratios were provided for the PRS, but no AUC, making the usefulness hard to evaluate.

- 4) The authors examine enrichment for AR binding sites, which is an interesting aspect of the paper. However, not much detail is given in the methods, making the analysis hard to evaluate. For example, what SNPs/loci were included in the analysis? Were all genome-wide significant SNPs in the analysis and/or the previous multi-ancestry meta-analysis included? Was any LD pruning done? What reference population was used for LDSC? What criteria or p-value threshold was used for assessing/including AR motif matches? Sometimes it is helpful to evaluate other binding sites for enrichment to ensure that the enrichment observed is truly for that particular motif and not seen with other motifs.
- 5) More detail is needed regarding the fine-mapping approach adopted in this paper. The authors state that they conducting fine mapping on all significant loci (presumably 171 loci), which resulted in 166 credible sets for the multi-ancestry analysis with 77% of sets containing less than 10 SNPs. The authors then compare the results to those seen with just the European analysis. However, they only provide a plot and p-value for the comparison with Europeans. They mention restricting the analysis to those that overlap, but do not provide any further information, making it difficult for the reader to fully evaluate this finding. The authors should provide more information about what the fine-mapping analysis results were for Europeans, so that the reader can understand the comparison and draw their own conclusions. Also, it would be helpful if the authors provided more detail about the fine-mapping method employed, such as how fine-mapping regions were determined, what size window was utilized, etc...
- 6) The authors mention that rs4704108 is an eqtl for ENC1, but it is not clear what dataset or criteria was used to determine this or even what the effect of the risk variant was on expression (e.g., increased or decreased expression).
- 7) Line 165. Instead of saying "331 potential causal variants", the author should say "331 potential functional variants". These variants just lead to increased risk and are not causal by themselves.

Reviewer #3 (Remarks to the Author): expert in prostate cancer androgen signalling

The authors have leveraged a large Japanese GWAS dataset and annotated the risk loci identified based on their enrichment with AR binding sites. This has yielded a novel polygenic risk score and nine novel loci including an association with a tumour suppressor gene, ZFH3. This has recently been found to be somatically mutated in multi-ethnic prostate cancer and is a candidate driver for prostate cancer tumorigenesis. The approaches used are not novel but are sound, the strength of the study lies in the findings and their potential translation into a new risk-associated diagnostic pathway.

RESPONSE TO REVIEWERS' COMMENTS

Reviewer #1 (Remarks to the Author): expert in prostate cancer genomics

The authors addressed the challenges of prediction of prostate cancer (PrCa) mortality risk by maximizing the statistical power of genetic data with trans-ancestry meta-analysis and focusing on binding sites of the androgen receptor (AR), which has a critical role in PrCa in a large Japanese cohort by conducting a trans-ethnic meta-analysis comprising more than 300,000 subjects. Their analyses identified 9 novel loci including ZFH3, a tumor suppressor gene. The authors stated that the trans-ancestry meta-analysis narrowed down these candidate causal variants, enriched in AR binding sites compared to European-only studies. A polygenic risk scores (PRS) analysis of candidate causal variants in AR binding sites showed among cancer-free subjects. Their results showed that individuals with a PRS in the top 10% have a higher risk of the future death of PrCa (HR: 5.57, $P = 4.2 \times 10^{-10}$).

This manuscript is of significance to prostate cancer epidemiology. . There is recent literature surrounding the calculation of a polygenic risk score in prostate cancer. . The novel component appears to relate to the method for finding significant variants specific to the androgen receptor (AR) using trans-ancestry meta-analysis in a large Japanese population.

Thank you very much for the positive comments.

Select related literature:

Chen, F.; Darst, B.F.; Madduri, R.K.; Rodriguez, A.A.; Sheng, X.; Rentsch, C.T.; Andrews, C.; Tang, W.; Kibel, A.S.; Plym, A., et al. Validation of a multi-ancestry polygenic risk score and age-specific risks of prostate cancer: A meta-analysis within diverse populations. *Elife* 2022, 11, doi:10.7554/eLife.78304.

Reference #14 in manuscript:

Conti DV, Darst BF, Moss LC, Saunders EJ, Sheng X, Chou A, Schumacher FR, Olama AAA, Benlloch S, Dadaev T, Brook MN, Sahimi A, Hoffmann TJ, Takahashi A, Matsuda K, Momozawa Y, Fujita M, Muir K, Lophatananon A, Wan P, et al. 2021. Trans-ancestry genome-wide association meta-analysis of prostate cancer identifies new susceptibility loci and informs genetic risk prediction. *Nature Genetics* 53:65–75.

DOI: <https://doi.org/10.1038/s41588-020-00748-0>, PMID: 33398198

Plym A, Penney KL, Kalia S, Kraft P, Conti DV, Haiman C, Mucci LA, Kibel AS. 2022. Evaluation of a multiethnic polygenic risk score model for prostate cancer. *Journal of the National Cancer Institute* 114:771–774. DOI: <https://doi.org/10.1093/jn93ci/djab058>, PMID: 337926

Thank you very much for pointing out the relevant papers to be cited. We incorporated these papers in the revised manuscript (Ref# 20, 14 and 19 in the revised manuscript).

I have a concern about this work supporting one of the conclusions of this manuscript. The authors claimed that their results support the androgen receptor (AR) plays fundamental and central roles not only PrCa susceptibility but on the future outcome of PrCa even in subjects without PrCa (Lines 247-248) and in "the potential early detection and therapeutic intervention for PrCa" (Line 104). This is in contradiction to Lines 281 to 284, where the authors stated that their analysis of AR binding sites in leukocytes, did not result in heritability enrichment of PrCa susceptibility. Thus, selecting prostate cancer tissue would be better to construct PRS. This finding decreases the utility of PRS in AR being used to predict PrCa in individuals who do not have PrCa. A better accompaniment to screening in PrCa prevention would be a blood test instead of an invasive biopsy of PrCa tissue which a clinician would not obtain a prostate tissue sample unless a patient had already been diagnosed or was being evaluated for PrCa.

Thank you very much for the comments. We are afraid that we made you confused about tissues as a source of AR binding sites in our analyses. Please excuse us for your confusion.

We obtained ChIP-seq data of AR in normal prostate available in public database. Using this data we could identify AR binding sites in normal prostate and as a result, we found strong enrichment of causal variants of PrCa in these AR binding sites.

We found reasonable enrichment of causal variants in AR binding sites in PrCa (not in normal prostate) which is also available in public database. However, the enrichment was much weaker in PrCa than in normal prostate, suggesting that AR

bindings sites in normal prostate available in public database would be useful as a source of causal variants of PrCa.

As the reviewer pointed out, AR binding sites in the leukocytes did not show enrichment for causal variants for PrCa – however, we showed that AR binding sites in normal prostate commonly available in public database showed the best fitness and we can use the data of AR binding. We are not arguing inter-individual differences in AR bindings in normal prostate tissues (we do not mean comparing AR bindings across prostate tissues from multiple healthy subjects to find the best enrichment).

Please note that we do not use AR binding sites in normal prostate from the patients analyzed (as discussed above, we are not focusing on inter-individual differences in AR-binding in different prostate tissues), suggesting that the AR binding sites information in normal prostate currently available is generalizable.

Thus, our results suggest that AR binding sites in normal prostate, even if not optimized for each patient, can contribute to prediction of development of PrCa in subjects without PrCa.

We modified the manuscript to make clear this point as follows in the Results.

(line 194~)

First, we evaluated the enrichment of heritability of PrCa susceptibility in AR binding sites of prostate tissues in common database. We computed LD scores using information of AR binding sites in prostate tissues obtained from the ChIP-atlas (this means that we used general AR binding in prostate and did not match origins of data between PrCa susceptibility and AR binding).

Additionally, I believe that the inclusion of a Japanese cohort to calculate polygenic risk scores lacks supportive documentation explaining why this population is an ideal population to study the genetics of prostate cancer. Why was the BBJ cohort selected? What is the incidence, morbidity and mortality of prostate cancer in Japanese men and how does it compare to African American men who carry a disproportionate share of prostate cancer burden. The authors should revise this manuscript to provide more documentation supporting their selection of this cohort.

Thank you very much for the comments. We agree that we should provide detailed information of the BBJ data and why we think that our findings would be generalizable.

Please note that we are not arguing Japanese population as ideal to analyze polygenic risk scores for PrCa. We chose the BBJ subjects based on data access to individual data and data availability of follow-up data. Additionally, under the view point of discovery of causal variants, trans-ethnic meta-analysis would be useful. Since previous studies recruiting a big case-control data in European population, non-European data would be useful to boost research of PrCa.

Please note that comparison between Japanese and other populations including African American is not a scope of the current study.

The epidemiology of PrCa in the Japanese is a little different between Asians and Europeans. In detail, prostate cancer incidence is approximately 45% lower in Asians compared with non-Hispanic whites (non-Hispanic whites are known to have lower PrCa incidence than African American). (US Cancer Statistics Working Group (June 2019). U.S. Cancer Statistics Data Visualizations Tool, based on November 2018 submission data (1999–2016) (US Department of Health and Human Services, Centers for Disease Control and Preventions and National Cancer Institute, accessed 1 September 2019); www.cdc.gov/cancer/dataviz).

In addition, the similarities of genetics underlying PrCa between EAS and EUR (and other populations) which were shown in the previous meta-analyses (Conti et al, Nat Genet 2021). In line with this, we found a strong genetic correlation between EUR and EAS ($\rho=0.88$).

Importantly, anti-AR therapy is a treatment option for PrCa regardless of populations (PMID: 33480983).

These support generalizability of the current findings while we should confirm it in future studies.

We modified the manuscript accordingly.

(line 100~)

Since European population is still the major source of genetic association studies, non-European population would be useful to find novel associations. While

Japanese has relatively low prevalence of PrCa in comparison with European populations and African Americans, the previous studies showed substantial genetic overlap among populations.

(line 122~)

As expected, we observed a strong genetic correlation of PrCa susceptibility between BBJ and Europeans (genetic effect correlation=0.88 and $p=0.36$ by popcorn software, indicating that genetic correlation is not different from 1, see Methods).

I do not detect any flaws in the data analysis. I believe the methodology is sound and meets the expected standards in this field of study. There appears to be enough detail provided in the methods to reproduce the work reported in this manuscript.

Thank you very much for the positive comments for our methods. We appreciate your high evaluation of our manuscript.

Reviewer #2 (Remarks to the Author): expertise in GWAS bioinformatics

This study conducts a meta-analysis of genome-wide association studies of prostate cancer, identifies 9 novel loci, creates a polygenic risk score (PRS) for prostate cancer and tests for association with prostate cancer death. Although the study is large and brings some insight into the genetic etiology of prostate cancer, it is limited by having only an incrementally larger dataset than the previous multi-ancestry GWAS and lacks an independent sample set for evaluating polygenic risk scores.

Thank you very much for your comments.

We agree with the reviewer that the data set is slightly larger than previous multi-ancestor GWAS. However, 9 novel signals are substantial number. Regarding polygenic risk scores, we evaluated death from prostate cancer in non-prostate cancer participants by using follow-up data. That is an uncommon trait and the advantage of our present study. In the revised manuscript, we added analyses in which we avoid potential overfitting to the test data set as written in the response to the 3rd comments.

Specific comments are below.

1) This study takes advantage of data from Biobank Japan (BBJ) to increase diversity; however, the added value is somewhat limited by the fact that the study excluded the East Asian cases and controls that were included in the previous multi-ancestry GWAS meta-analysis from prostate cancer. Thus, the number of EAS is about the same, although the number of EAS controls is larger. The authors excluded the EAS subjects from the previous study, because some of them overlapped with BBJ. This is a reasonable approach; however, the authors could have excluded the Biobank Japan subjects from the EAS summary statistics of the previous GWAS using the MetaSubtract R package and then included the non-overlapping subjects in the analysis.

Thank you very much for your comments. Please note that summary statistics specific to EAS in the previous study is not publicly available.

After negotiation, we obtained the EAS summary statistics of the previous study and conducted subtraction of the BBJ previous GWAS from the EAS summary statistics.

We excluded the BBJ previous GWAS based on inverse variance weighted method (since MetaSubtract R package requires detailed information in the summary statistics which were not available). Then we performed trans-ethnic meta-analysis including non-overlapping subjects. As a result, however, we observed rather decreased number of novel signals (from 9 to 8). GINS1 in chr 20 with a significant association in the original submission did not hold its significant association. Importantly, this locus is supported as significant locus in the upcoming trans-ethnic meta-analysis (Wang et al, revision submitted to Nature Genetics), suggesting that this locus is a true signal.

Thus, since we did not increase the number of novel signals, we would like to keep the original results of our manuscript.

We put a figure of Manhattan plot below in the trans-ethnic meta-analysis in which we subtracted the BBJ stats from EAS stats and meta-analyzed with the latest BBJ stats and non-EAS sumstats (this was not shown in the revised manuscript). This Manhattan plot did not show the GINS1 peak in chr 20.

2) The author identified 171 independent loci (including 9 novel loci). However, no criteria are given as to how the authors determined that the loci were independent. The analysis method used is also not clear. The authors mention using MANTRA in the methods, meta-analysis assuming random effects on line 129 and fixed effects meta-analysis on line 132. In addition, instead of adopting the standard genome-wide p-value threshold of $p < 5 \times 10^{-8}$, the authors use a slightly different significance

threshold based on \log_{10} Bayes Factor > 6 and fixed effects P-value $< 1 \times 10^{-5}$. In other studies that have employed these criteria, the loci have not always replicated in subsequent studies. Most of the novel loci that the authors report (except for one locus) have a fixed effects meta-analysis p-value $< 5 \times 10^{-8}$. Given that the one novel locus with a fixed effects $p > 5 \times 10^{-8}$, is driven primarily by BBJ (with little or no association in other ancestries), it does make one question whether this is indeed a real locus. Are there other signals in the region supporting this association in LocusZoom plots? Are the findings replicated in another study, such as in the EAS subjects from the previous multi-ancestry analysis that were not included in this study? In the absence of replication, maybe it would be prudent to classify this locus as suggestive (not genome-wide significant).

Thank you very much for the comments.

Regarding the description of independent signals, please excuse us for making you confused.

We defined a significantly associated locus of a lead variant as 1 Mb of its surrounding sequences in both directions. Then, we extended the region to nearby significant variants and their 1 Mb surrounding sequences as far as a significant variant was contained in the defined region.

We modified the description in the Methods accordingly.

(line 410~)

For the trans-ethnic meta-analysis by using METAL, we defined a significantly associated locus of a lead variant as 1 Mb of its surrounding sequences in both directions. Then, we extended the region to nearby significant variants and their 1 Mb surrounding sequences as far as a significant variant was contained in the defined region.

Regarding statistical significance, we adopted the strategy to set statistical significance based on both fixed and random effects in accordance to the previous studies (PMID: 23406875 and 33020668). Regarding the significant locus not exceeding the p-value of 5×10^{-8} in the fixed model, thank you very much for pointing this out. We agree that we should treat this variant with some caution since only this 'lead' variant in the novel loci did not exceed the p-value of 5×10^{-8} in the fixed model.

However, other variants in this region showed associations with p-value smaller than 5×10^{-8} . Thus, all of the nine regions exceeded the p-value of 5×10^{-8} in the fixed model. (In the original submission, the variant exceeding 5×10^{-8} in the ZFH3 was not featured in the Table since we prioritized BF in the random effect and picked up the variant with highest BF in each region.)

In the revised manuscript, we added the LocusZoom of the locus putting stress on $P < 5 \times 10^{-8}$ in the fixed model as below (Fig S3). We also make clear in the footnote of the Table to describe the significant association in the fixed model in this region.

Fig. S3. A locus plot for the *ZFH3* region.

While we prioritized rs8052683 in Table 1 based on BF, this region contains variants exceeding p-value of 5×10^{-8} in the fixed model.

Table 1. The novel significant loci for prostate cancer in trans-ethnic meta-analysis.

rsID	Chr	Position	Gene	Location	Ref	Alt	META				BBJ			EUR		
							Beta	BF	Fixed P	Freq	Beta	P value	Freq	Beta	P value	Freq
rs2235558	1	24783311	NIPAL3	intronic	A	G	0.039	6.408	1.71E-08	0.747	0.064	8.63E-04	0.618	0.032	5.01E-05	0.747
rs4893909	2	181269224	CWC22/SCHLAP1	intergenic	T	G	0.043	7.110	3.10E-09	0.672	0.061	5.91E-04	0.729	0.043	1.43E-06	0.550
rs4704108	5	73392666	ARHGEF28/LINC01335	intergenic	G	A	-0.045	7.172	2.41E-09	0.214	-0.086	1.69E-05	0.267	-0.038	2.05E-05	0.383
rs35055448	8	81135961	TPD52/MIR5708	intergenic	G	A	-0.041	6.072	4.87E-08	0.543	-0.064	1.74E-04	0.252	-0.034	1.16E-04	0.153
rs11259192	10	5726821	FAM208B	5' UTR	G	A	0.062	7.217	3.24E-09	0.069	0.086	8.13E-03	0.119	0.054	2.38E-06	0.025
rs11857866	15	33349563	FMN1	intronic	T	C	0.043	7.282	2.47E-09	0.201	0.065	1.66E-03	0.310	0.039	2.33E-06	0.262
rs8052683	16	73002421	ZFX3 *	intronic	G	A	-0.038	7.086	1.85E-07	0.253	-0.113	4.46E-09	0.296	-0.026	3.05E-03	0.185
rs11087515	20	25399979	GINS1	intronic	G	GA	0.039	6.056	3.83E-08	0.105	0.054	6.09E-02	0.477	0.034	1.68E-05	0.470
rs10154043	21	35535103	MRPS6/LINC00310	intergenic	C	T	-0.038	6.558	1.27E-08	0.229	-0.047	1.66E-02	0.441	-0.037	1.56E-06	0.536

Chr, chromosome; Ref, reference allele; Alt, alternative allele; META, meta-analysis; BBJ, Biobank Japan; EUR, Europeans; AFR, Africans; HIS, Hispanic, Beta; beta of alternative allele; BF, bayse factor; Fixed P, P value of fixed-effect meta-analysis; Freq, alternative allele frequency; UTR, untranslated region. Prioritized variants based on BAF are indicated *; Other variants in this gene region showed significant associations with p-value smaller than 5E-08 in the fixed effect model (Fig S3).

3) The authors test the association between the PRS and the risk of prostate cancer death among a cohort of men who do not have prostate cancer at baseline in BBJ. However, this analysis is problematic in that the authors appear to have used the same men included in the discovery and creation of PRS to test for the association with prostate cancer mortality. Although one may argue that the endpoint here is prostate cancer mortality and not incidence, in order to die from prostate cancer, one must first be diagnosed with prostate cancer. The men who die are just a subset of those who were diagnosed with prostate cancer. They are not an independent set of men, and mortality is not independent outcome. Moreover, the more important question to ask is whether the PRS is predictive of death among men who have already been diagnosed with prostate cancer. Clinically, this is more useful than predicting prostate cancer death among those who do not have prostate cancer yet. Other prostate PRS have been shown to predict prostate cancer death in a cohort of healthy men, because prostate PRS are predictive of prostate cancer incidence (which is necessary for prostate cancer death). Finally, it is not clear which PRS are presented in the paper. The authors discuss testing several LD pruning thresholds in the methods and investigating the integration of functional annotations, but do not state which LD threshold was ultimately used OR provide the variants/weights used in either PRS presented (e.g., the PRS without inclusion of AR information and the PRS with inclusion of AR information). Both PRS were also only tested in EAS, so

their performance in other ancestries is unknown. Hazard ratios were provided for the PRS, but no AUC, making the usefulness hard to evaluate.

Thank you very much for the comments.

We are afraid that we made you confused about our study design of PRS. We assume that the reviewer concerns about potential overfitting. If we computed PRS for the case subjects used for the case-control studies and subjected them to the survival analyses, we agree that the analyses should result in potential overfitting.

On this point, we computed PRS for the control samples (not cases) in the association studies and subjected them to the survival analyses. These control samples do not contribute to potential inflation of the estimate of beta coefficients in the case-control study (rather potentially deflate the effect sizes since these subjects are potential cases). Thus, there is a limited possibility of potential overfitting.

To confirm this, we conducted additional analyses in the revised manuscript. We split the control subjects in the original submission into two (1.subjects only for the case-control association study and 2.subjects only for the survival analysis) to avoid sample overlapping as follows (Fig. S4 in the revision).

As a result, we observed consistent results (PRS based on AR binding sites and lead variants outperforming over normal PRS) in the future mortality risk for PrCa in the control subjects. We added the results to Supplementary materials.

Supplementary Table 12. Cox proportional hazard model for death from prostate cancer using polygenic risk scores (PRSs) without any sample overlap between case-control and survival studies.

	score	HR (95%CI)	Pr(> z)
GWAS_PRS	quantitative	1.44 (1.24-1.67)	1.7x10 ⁻⁷
	Top 10% vs bottom 50%	4.01 (2.40-6.70)	1.1x10 ⁻⁵
	Top 1% vs bottom 50%	6.61 (2.30-19.04)	4.6x10 ⁻⁴
AR_prioritized_PRS	quantitative	2.08 (1.62-2.67)	9.3x10 ⁻⁹
	Top 10% vs bottom 50%	4.58 (2.66-7.89)	4.0x10 ⁻⁸
	Top 1% vs bottom 50%	16.70 (2.66-7.89)	8.3x10 ⁻¹²

Regarding whether the PRS could predictive the death in patients with PrCa, while this is not the scope of this study since this analysis is not optimized for disease course of PrCa, we additionally analyzed data for associations between PrCa PRS (based on AR binding sites and lead variants) and mortality in the PrCa cases (not controls). As a result, while not reaching the statistical significance due to limited power, we observed the trend of positive associations. We added the results to the revised manuscript.

Supplementary Table 13. Cox proportional hazard model for death in subjects with prostate cancer using AR-prioritized polygenic risk scores (PRSs).

score	HR (95%CI)	Pr(> z)
Top 10% vs bottom 50%	1.08 (0.89-1.30)	0.45
Top 1% vs bottom 50%	1.37 (0.81-2.33)	0.25

(line 266~)

We confirmed consistent results of better fitness of AR-informed PRS over normal PRS in additional analyses in which we avoid sample overlap between survival analyses and case-control studies (Supplementary Table S12 and Methods). In addition, we observed a trend of positive associations between AR-informed PRS

and mortality in PrCa subjects (Supplementary Table S13).

The strong association with PrCa death in all subjects rather than PrCa death in patients with PrCa is reasonable since this GWAS is case-control GWAS and not optimized for associations with disease course in PrCa.

We are sorry for making you confused about the final model of PRS. We used PRS with r^2 of 0.6 and p value threshold of 5×10^{-4} , containing 1,107 SNPs. We made clear this point in the revised manuscript.

(line 250~)

As a result, the model with r^2 of 0.6 and p value threshold of 5×10^{-4} , containing 1,107 SNPs, reached the highest AUROC (0.686; 95% CI, 0.676–0.695).

We applied the PRS to ask associations with future PrCa death in subjects not developed PrCa at registry (controls). Since only a fraction of subjects in controls would die of PrCa in the follow-up period and the follow-up data contains many censored data to be taken into consideration (and not properly handled in ROC), we believe that HR is appropriate to estimate the associations in our data set. However, we appreciate future studies to estimate PRS in more detail. We added a description about possible generalization of the PRS and future studies to confirm the PRS to the Discussion.

(line 349~)

Third, prediction of PRS on mortality due to PrCa in cancer-free subjects and cancer cases should be addressed in European populations to show its generalizability, especially in a cohort specific to PrCa.

4) The authors examine enrichment for AR binding sites, which is an interesting aspect of the paper. However, not much detail is given in the methods, making the analysis hard to evaluate. For example, what SNPs/loci were included in the analysis? Were all genome-wide significant SNPs in the analysis and/or the previous multi-ancestry meta-analysis included? Was any LD pruning done? What reference population was used for LDSC? What criteria or p-value threshold was used for assessing/including AR motif matches? Sometimes it is helpful to evaluate other

binding sites for enrichment to ensure that the enrichment observed is truly for that particular motif and not seen with other motifs.

Thank you very much for the comments. We agree that we should provide more details of analyses for enrichment of AR binding sites.

We checked AR binding motif only for lead variants to show an example of a risk variant altering AR binding motif (namely, ZFH3). Please note that we did not conduct AR motif matches for all variants in GWAS. Since AR binding is different from tissue to tissue, we used CHIP seq data rather than motif.

We evaluated enrichment of PRCa-susceptibility signals in AR bindings sites in normal prostate tissue in different two aspects, namely, polygenic signals by LDSC and putative causal variants by GREGOR. In either method, we did not conduct any pruning, since both LDSC and GREGOR take LD into consideration.

For the LDSC, we used all variants in the GWAS except the MHC region (in total, around 6.7M variants) to evaluate enrichment of polygenic signals in the AR binding sites (please note that this means that we did not select or filter variants – practically LDSC takes 1kg or hapmap variants for the analyses since these variants are highly accurately imputed and result in stable and trustable associations). Since LDSC takes care of polygenic signals, the associations were not changed with or without GWAS significant variants. Taking CHIP seq data of AR in normal prostate, we computed LD scores of AR binding in normal prostate using LD structure of East Asians and Europeans, respectively, and used them for LDSC. We used the BBJ sumstats with use of EAS LD scores and European population sumstats with European's LD scores for LDSC. LDSC takes LD into consideration (LD info is captured by LD scores).

For enrichment of putative causal variants, we used three sets of variants, namely, 1.all GWAS significant lead signals in the meta-analyses (171 variants) 2.all putative causal variants (331 variants) with $PPI > 0.1$ and 3.all putative causal variants with $PPI > 0.5$ (96 variants).

We evaluated enrichment of these sets of variants in the AR bind sites by GREGOR. Since GREGOR takes care of LD between lead variants and other variants to compute enrichment of list of variants on bed files specified (in our analyses, AR

binding sites), we take LD structure into consideration. For this analysis, we used LD structures in European population since a large part of meta-analyses subjects were from Europeans.

In addition, we added the following sentences to describe the details.

(line 455~)

For LDSC, we assessed heritability enrichment in AR binding sites as previously described for each of European and BBJ GWAS results. Taking ChIP seq data of AR in normal prostate, we computed LD scores of AR binding in normal prostate using LD structure of East Asians and Europeans, respectively, and used them for LDSC. We used the BBJ sumstats with use of EAS LD scores and European population sumstats with European's LD scores for LDSC. We used the 53 basic model annotations (v1.0) to control inflation of the results as previously described. We excluded variants within the major histocompatibility complex (MHC) region (chromosome 6: 25-34 Mb) from the regression analysis.

For GREGOR, we calculated the fold-enrichment expectation and an enrichment P-value that represents the probability that the overlap of control SNPs represented as a cumulative probability distribution is greater than or equal to the observed overlap with PrCa potential causal variants. We conducted the analysis using three SNP sets: lead variants identified in the meta-analysis, the variants with PP > 0.1, the variants with PP > 0.5. Reference based on European population was used in the analysis because the majority of the samples are from Europeans.

While we did not conduct motif search, when we took FOXA1 ChIP seq data in normal prostate, we did not observe heritability enrichment of PrCa susceptibility for the FOXA1 binding sites in normal prostate ($p=0.075$).

This point was also added to the Results section.

(line 204~)

In contrast, FOXA1 binding in prostate obtained from ChIP-atlas as control did not show heritability enrichment ($p=0.075$).

5) More detail is needed regarding the fine-mapping approach adopted in this paper. The authors state that they conducting fine mapping on all significant loci

(presumably 171 loci), which resulted in 166 credible sets for the multi-ancestry analysis with 77% of sets containing less than 10 SNPs. The authors then compare the results to those seen with just the European analysis. However, they only provide a plot and p-value for the comparison with Europeans. They mention restricting the analysis to those that overlap, but do not provide any further information, making it difficult for the reader to fully evaluate this finding. The authors should provide more information about what the fine-mapping analysis results were for Europeans, so that the reader can understand the comparison and draw their own conclusions. Also, it would be helpful if the authors provided more detail about the fine-mapping method employed, such as how fine-mapping regions were determined, what size window was utilized, etc...

We thank the reviewer for the valuable comments. We totally agree that we should provide more details of fine-mapping methods.

Regarding the fine-mapping methods, we adopted Wakefield's asymptotic Bayes factors (ABF) derivation in which ABF is computed based on beta of variants, SE of variants and Z scores of associations. This is a LD-independent method and enables us to compare the fine-mapping results across populations. We defined each association region based on lead variants and distance from lead variants, namely, 1 Mb of surrounding sequences centering lead variants in both directions. If another significant variant was within 1 Mb from the region, we extended the region to the nearby significant variants and its surrounding region. This step was continued until any other significant variants are not located within 1 Mb from the defined region. Then, we can compute PPI of each variant in the defined each region, based on ABF. Accordingly, we can define 95% credible set in each region based on PPI.

Since the number of variants is different between meta-analysis and European GWAS, we restricted variants in each locus common in the two data sets. This strategy enables us to fairly compare the number of variants in 95% credible sets between the results.

We added the sentences to describe the details in the Methods section.

(line 410~)

For the trans-ethnic meta-analysis by using METAL, we defined a significantly

associated locus of a lead variant as 1 Mb of its surrounding sequences in both directions. Then, we extended the region to nearby significant variants and their 1 Mb surrounding sequences as far as a significant variant was contained in the defined region. For each locus, we calculated asymptotic Bayes factors as previously described, which is a LD-independent method. Bayes factors can be approximated from summary statistics (such as p-value and standard error of the effect size of each variant from GWAS) without individual-level genotype data. Then we defined the subset of SNPs based on posterior probability (PP)PP, as 95% likely to contain the causal disease-associated SNP (<https://github.com/chr1swallace/finemap-psa>). These credible SNP sets were then annotated for putative function.

We further compared the number of variants in the credible sets between the results of the trans-ethnic meta-analysis and that of European alone. We restricted variants present in both the trans-ethnic meta-analysis and European results to harmonize conditions and conducted fine-mapping as described above for each of the trans-ethnic meta-analysis and European alone. Then we focused on the significant regions overlapped between the two results and compared the number of variants in the credible sets. The comparison was estimated by paired t-test using R (version 4.0.3).

6) The authors mention that rs4704108 is an eQTL for ENC1, but it is not clear what dataset or criteria was used to determine this or even what the effect of the risk variant was on expression (e.g., increased or decreased expression).

Thank you very much for the comments. We agree that we should provide more details about eQTL information of ENC1. We used eQTL data in prostate tissues in the GTEx. In the revised manuscript, we provide the details as follows.

(line 143~)

rs4704108, located at the intergenic region between ARHGEF28 and LINC01334, is an eQTL for ENC1 in prostate tissues in the GTEx data³⁰ and in high LD with the lead eQTL SNP (rs17636369) of ENC1, suggesting that rs4704108 (or its tightly linked variant) is associated with PrCa via altering expression of ENC1 in the prostate. Risk allele of rs4704108 decreases expression of ENC1.

7) Line 165. Instead of saying “331 potential causal variants”, the author should say “331 potential functional variants”. These variants just lead to increased risk and are not causal by themselves.

Thank you for the comments. We agree that if the variants are in AR binding sites, “potential functional variants” is appropriate.

At the same time, since we statistically finemapped the variants and defined the set of 331 variants with possibility of being causal ($PPI > 0.1$). This PPI does not directly link to functionality since not all of them were in AR binding sites. Thus, we are afraid that “331 potential causal variants” is appropriate.

Reviewer #3 (Remarks to the Author): expert in prostate cancer androgen signalling

The authors have leveraged a large Japanese GWAS dataset and annotated the risk loci identified based on their enrichment with AR binding sites. This has yielded a novel polygenic risk score and nine novel loci including an association with a tumour suppressor gene, ZFX3. This has recently been found to be somatically mutated in multi-ethnic prostate cancer and is a candidate driver for prostate cancer tumorigenesis. The approaches used are not novel but are sound, the strength of the study lies in the findings and their potential translation into a new risk-associated diagnostic pathway.

We are very happy to these comments.

Thank you very much for your comments and highly evaluating our manuscript.

REVIEWERS' COMMENTS

Reviewer #1 (Remarks to the Author):

Thank you for addressing my concerns in your manuscript. I believe the authors have fully addressed my comments and the manuscript is now improved with added clarity. I do not have any additional comments.

Reviewer #2 (Remarks to the Author):

The authors have addressed most comments. However, the authors should acknowledge in the discussion that they were unable to evaluate the association between the PRS and progression and death among men diagnosed with prostate cancer. This is a major limitation of this study. The authors should also acknowledge the study by Meisner et al that previously demonstrated that prostate PRS was predicted of prostate cancer death in the UK Biobank (AJHG 2020). Finally, I respectfully disagree with the authors about the use of the term "causal variants". The vast majority of SNPs discovered from GWAS are neither necessary or sufficient by themselves to cause disease. Thus, they do not meet the standard biological definition of causality. Having a PPI>0.1 is just a statistical measure of likelihood of the SNP being related in some manner to disease.

RESPONSE TO REVIEWERS' COMMENTS

Reviewer #1 (Remarks to the Author):

Thank you for addressing my concerns in your manuscript. I believe the authors have fully addressed my comments and the manuscript is now improved with added clarity. I do not have any additional comments.

Thank you very much for the positive comments.

Reviewer #2 (Remarks to the Author):

The authors have addressed most comments.

Thank you very much for the positive comments.

However, the authors should acknowledge in the discussion that they were unable to evaluate the association between the PRS and progression and death among men diagnosed with prostate cancer. This is a major limitation of this study.

Thank you for the comments. We put an additional sentence about this point.

The authors should also acknowledge the study by Meisner et al that previously demonstrated that prostate PRS was predicted of prostate cancer death in the UK Biobank (AJHG 2020).

Thank you for the comments. We refer this paper (Meisner et al, AJHG 2020) in the introduction and results sections of the revised manuscript.

Finally, I respectfully disagree with the authors about the use of the term "causal variants". The vast majority of SNPs discovered from GWAS are neither necessary or sufficient by themselves to cause disease. Thus, they do not meet the standard biological definition of causality. Having a PPI>0.1 is just a statistical measure of likelihood of the SNP being related in some manner to disease.

Thank you for the comments. We noticed that the main text still contained several assertive expression of "causal variants". Since we agree that statistical finemapping only helps us to find candidates of causal variants to be functionally tested, we modified revised manuscript to term "statistically finemapped variants" or "candidate/putative/potentially causal variants" throughout the paper.